# Mechanistic Insights on Salicylic Acid Mediated Enhancement of Photosystem II Function in Oregano Seedlings Subjected to Moderate Drought Stress [note 1]

**DOI:** 10.3390/plants12030518

**Published:** 2023-01-23

**Authors:** Michael Moustakas, Ilektra Sperdouli, Julietta Moustaka, Begüm Şaş, Sumrunaz İşgören, Fermín Morales

**Affiliations:** 1Department of Botany, Aristotle University of Thessaloniki, 54124 Thessaloniki, Greece; 2Institute of Plant Breeding and Genetic Resources, Hellenic Agricultural Organisation–Demeter (ELGO-Demeter), 57001 Thessaloniki, Greece; 3Department of Plant and Environmental Sciences, University of Copenhagen, Thorvaldsensvej 40, 1871 Frederiksberg C, Denmark; 4Instituto de Agrobiotecnología (IdAB), CSIC-Gobierno de Navarra, Avda. de Pamplona 123, 31192 Navarra, Spain

**Keywords:** chlorophyll fluorescence, photosynthetic efficiency, light reactions, excess excitation energy, *Origanum vulgare*, reactive oxygen species, photoinhibition, photochemistry, photoprotection

## Abstract

Dramatic climate change has led to an increase in the intensity and frequency of drought episodes and, together with the high light conditions of the Mediterranean area, detrimentally influences crop production. Salicylic acid (SA) has been shown to supress phototoxicity, offering photosystem II (PSII) photoprotection. In the current study, we attempted to reveal the mechanism by which SA is improving PSII efficiency in oregano seedlings under moderate drought stress (MoDS). Foliar application of SA decreased chlorophyll content under normal growth conditions, but under MoDS increased chlorophyll content, compared to H_2_O-sprayed oregano seedlings. SA improved the PSII efficiency of oregano seedlings under normal growth conditions at high light (HL), and under MoDS, at both low light (LL) and HL. The mechanism by which, under normal growth conditions and HL, SA sprayed oregano seedlings compared to H_2_O-sprayed exhibited a more efficient PSII photochemistry, was the increased (17%) fraction of open PSII reaction centers (q*p*), and the increased (7%) efficiency of these open reaction centers (F*v*′/F*m*′), which resulted in an enhanced (24%) electron transport rate (ETR). SA application under MoDS, by modulating chlorophyll content, resulted in optimized antenna size and enhanced effective quantum yield of PSII photochemistry (Φ*_PSII_*) under both LL (7%) and HL (25%), compared to non-SA-sprayed oregano seedlings. This increased effective quantum yield of PSII photochemistry (Φ*_PSII_*) was due to the enhanced efficiency of the oxygen evolving complex (OEC), and the increased fraction of open PSII reaction centers (q*p*), which resulted in an increased electron transport rate (ETR) and a lower amount of singlet oxygen (^1^O_2_) production with less excess excitation energy (EXC).

## 1. Introduction

Drought is the prevailing environmental factor affecting several physiological and biochemical processes of plants that detrimentally influences global crop production [1,2,3,4]. Drought stress (DS) episodes are expected to increase in frequency, intensity, and duration as a result of climate change [5,6]. Drought affects plant performance at practically every stage, from seed germination up to the growth and development of the adult plant [7]. DS hampers plants’ cell division, elongation, and differentiation, disturbs osmotic homeostasis, initiating turgor loss, impairs photosynthetic function, disturbing light energy balance, and eventually decreases plant productivity [7,8,9,10]. Plants must maintain an equilibrium between light energy capture and its use in photochemistry, which is altered under DS [2,6,11,12]. Under DS, plants close their stomata to decrease H_2_O loss, which results in lesser CO_2_ entrance into the leaf and lower CO_2_ fixation [13,14,15]. Consequently, under DS, the absorbed light energy exceeds chloroplasts’ capabilities’ use, and thus the photosynthetic apparatus, and particularly photosystem II (PSII), is exposed to this overdose of absorbed light energy [16,17,18,19]. This overdose of absorbed light energy, due to the reduction in photochemical energy use, must be dissipated as heat in order to prevent the formation of reactive oxygen species (ROS) [20,21,22]. The photoprotective mechanism that prevents ROS formation is the process of non-photochemical quenching (NPQ) [22,23,24,25,26,27], which leads to a decrease in the electron transport rate (ETR) [28,29].

Sunlight absorption by light-harvesting pigment-protein complexes (LHCs) results in singlet-state excitation of a chlorophyll *a* molecule (^1^Chl*), which can be de-excited and return to the ground state by several pathways; it can be re-emitted as chlorophyll fluorescence, it can be transferred to reaction centres to drive photosynthetic electron transport (photochemistry), it can be de-excited by thermal dissipation, which results in NPQ of chlorophyll fluorescence, or it can decay via the triplet state (^3^Chl*), the basal dissipation. Triplet-state chlorophylls (^3^Chl*) are created from ^1^Chl* through intersystem crossing [23,25,27,30,31]. With increasing light intensity, there is a decrease in the efficiency of use of excitons in photochemistry, and an increase in NPQ [23,25,27,30,31].

In the light reactions of photosynthesis, ROS, such as superoxide anion radical (O_2_**^•^^−^**), hydrogen peroxide (H_2_O_2_), and singlet oxygen (^1^O_2_), are constantly formed at basal levels, but retained in homeostasis by non-enzymatic and enzymatic antioxidants [30,31,32,33]. Drought stress breaks the equilibrium between the formation and removal of ROS in plants [6], and ROS formation increases exceptionally [34], triggering oxidative stress that causes membrane damage, degradation of proteins and inactivation of enzymes, resulting in damaged cellular components [35,36,37]. Thus, during DS, despite the existence of photoprotective mechanisms, the increased production of ROS leads to photooxidative damage in plant tissues [38,39,40,41,42]. ROS-induced damage in plant tissues is one of the major factors that limit plant growth under drought stress [43,44]. 

Stomatal closure under DS limits CO_2_ availability, which results in reduced photochemical light energy use with the consequence of diverging electrons from the electron transport chain to molecular oxygen, generating the superoxide anion radical (O_2_^•–^) at the end of PS I [34]. Simultaneously, energy transfer from the excited triplet state of PSII reaction centre chlorophyll *a* (^3^P680*), and even from antenna chlorophylls in their triplet states (^3^Chl*), to molecular oxygen generates ^1^O_2_ which harms thylakoid membranes and can further produce the other ROS, O_2_**^•^^−^** and H_2_O_2_ [20,33,38,45,46].

Drought stress, with the concurrent action of high sunlight irradiance under natural conditions in the Mediterranean area, may become a severe threat to crop production [28,47,48,49,50]. Under such conditions of DS and high light irradiance, enormous ROS production happens [49,50,51,52]. However, in DS seedlings, there is the possibility of down-regulating chlorophyll synthesis and downscaling the light-harvesting complexes of PSII; this will thus adapt plants not to absorb excess light, thereby reducing ROS production [53]. Plants with a smaller antenna size and lower chlorophyll conte absorb less light energy, which results in lower ROS generation [54]. Thus, reduced leaf chlorophyll content has been proposed as a possible method to decrease sunlight absorption and improve photosynthetic function by reducing photooxidative stress, especially under the high light conditions of Mediterranean climates [54,55,56,57,58,59,60]. 

Salicylic acid (SA), which belongs to the diverse group of phenolics, is an important plant hormone interrelated with the other plant hormones and performs a significant role in plant stress defense against biotic or abiotic stresses [61,62]. An amplified SA production occurs through induction of plant defense responses with a simultaneous decrease in auxin biosynthesis, and their concurrent action orchestrates synchronized defense and plant growth responses [61,62]. SA has been reported to ameliorate the unfavorable consequences of DS and salinity, acting as a growth regulator and an antioxidant, improving the osmotic potential, transpiration rate, stomatal conductance, biochemical parameters, repairing membrane injury and restoring photosynthetic function and nutrient uptake [63,64,65].

Salicylic acid’s impact on plants cannot be globalized, as the influence may vary not only with the concentration and the method of application, but also with the plant species and the exposure duration [60,66]. Foliar application of SA in tomato plants suppressed phototoxicity by decreasing chlorophyll content and offering photoprotection of PSII [60]. Thus, SA application was suggested to improve PSII function by reducing photoinhibition and photodamage [60,67]. Plant productivity is described by the photochemical efficacy of the absorbed amount of light energy [68]. Breeding for improved photosynthesis and light energy use in crops is a manageable and a useful shorter-term addition to genetic engineering to enhance crop potential [69].

*Origanum vulgare* L. is a perennial flowering species in the family *Lamiaceae*, native to the Mediterranean region and Central Asia and widely used both as a medicinal and culinary herb, especially in the Greek, Italian, Turkish, Mexican, Spanish, and French cuisine. The objectives of this study were to characterize the functional differences in photosystem II (PSII) of oregano (*Origanum vulgare* L.) seedlings, with or without foliar application of 1 mM salicylic acid (SA), grown under optimal conditions or under moderate drought stress (DS). In addition, we aimed to determine the molecular mechanisms in the allocation of the absorbed light energy in PSII of oregano seedlings sprayed with SA, under DS and low light (LL), or DS and high light (HL), and to elucidate the mechanism by which SA improves PSII efficiency under DS.

## 2. Results

### 2.1. Chlorophyll Content and Maximum Efficiency of Photosystem II under Normal Growth and Moderate Drought Stress 

Leaves of oregano seedlings grown under optimal conditions were sprayed with 1 mM SA or double distilled H_2_O (control), and 72 h after spraying, the chlorophyll content was assessed. While chlorophyll content decreased significantly in the SA-sprayed oregano leaves under optimal growth conditions, compared to control (H_2_O-sprayed) (Figure 1a), the maximum efficiency of PSII photochemistry (F*v*/F*m*) remained unchanged after SA treatment (Figure 1b). 

Under moderate drought stress (MoDS) chlorophyll content, decreased significantly in both H_2_O-sprayed (−47%) and SA-sprayed leaves (−32%), compared to H_2_O-sprayed non-stressed leaves (control). Thus, chlorophyll content remained higher in SA-sprayed leaves compared to H_2_O-sprayed leaves (Figure 1a). F*v*/F*m* decreased significantly in MoDS H_2_O-sprayed oregano leaves compared to both non-stressed H_2_O-sprayed (−4%) and SA-sprayed leaves (−4%) (Figure 1b). Under MoDS, SA-sprayed leaves exhibited higher F*v*/F*m* values (2%) compared to H_2_O-sprayed MoDS leaves, but significantly lower values (−3%) compared to non-stressed SA-sprayed leaves. 

### 2.2. Allocation of Absorbed Light Energy in Photosystem II under Normal Growth and Moderate Drought Stress

The light energy distribution to photochemistry (Φ*_PSII_*), photoprotective heat dissipation (Φ*_NPQ_*), and non-regulated energy loss (Φ*_NO_*), was estimated under optimal growth conditions and MoDS, in both H_2_O-sprayed and SA-sprayed leaves.

The effective quantum yield of PSII photochemistry (Φ*_PSII_*), under optimal growth conditions, did not differ between H_2_O-sprayed and SA-sprayed leaves at low light intensity (LL, 205 μmol photons m^−2^ s^−1^, equal to the growth light intensity) (Figure 2a). Under MoDS and LL, SA-sprayed leaves displayed significantly higher (7%) effective quantum yield of PSII photochemistry (Φ*_PSII_*), compared to H_2_O-sprayed leaves (Figure 2a). However, under high light intensity (HL, 1000 μmol photons m^−2^ s^−1^), SA-sprayed leaves of oregano seedlings presented a significantly higher quantum yield of PSII photochemistry (Φ*_PSII_*) under both optimal conditions (25%) and under MoDS (25%) compared to H_2_O-sprayed leaves (Figure 2a).

The quantum yield of regulated non-photochemical energy loss (Φ*_NPQ_*), under normal growth conditions or MoDS, did not differ between H_2_O-sprayed and SA-sprayed leaves, at LL (Figure 2b). However, under HL, SA-sprayed leaves of oregano seedlings had significantly lower heat dissipation (Φ*_NPQ_*), under both optimal conditions (−11%) and under MoDS (−3%), compared to H_2_O-sprayed leaves (Figure 2b).

The quantum yield of non-regulated energy loss (Φ*_NO_*), under optimal growth conditions, did not differ between H_2_O-sprayed and SA-sprayed leaves, at both LL and HL (Figure 3a), while under MoDS, SA-sprayed leaves displayed significantly lower Φ*_NO_* at both LL (−15%) and HL (−8%) compared to H_2_O-sprayed leaves (Figure 3a).

### 2.3. Changes in the Redox State of the Plastoquinone Pool, the Electron Transport Rate, and the Efficiency of Open Photosystem II Reaction Centers under Normal Growth and Moderate Drought Stress

The fraction of open PSII reaction centers (q*p*), representing the redox state of quinone A (Q*_A_*) under optimal growth conditions at LL, did not differ between H_2_O-sprayed and SA-sprayed leaves; however, at HL, SA-sprayed leaves had a higher fraction of open PSII reaction centers (17%) (Figure 3b). Under MoDS, SA-sprayed leaves retained a higher fraction of open PSII reaction centers, at both LL (9%) and HL (23%) (Figure 3b).

The electron transport rate (ETR), under optimal growth conditions, did not differ between H_2_O-sprayed and SA-sprayed leaves at LL (Figure 4a), while under MoDS, SA-sprayed leaves displayed a significantly higher ETR (7%) compared to H_2_O-sprayed leaves (Figure 4a). Under HL, SA-sprayed leaves of oregano seedlings presented a significantly higher ETR, under both optimal conditions (24%) or under MoDS (25%), compared to H_2_O-sprayed leaves (Figure 4a). 

Τhe efficiency of excitation energy capture by the open PSII rection centers (F*v*′/F*m*′) under optimal growth conditions at LL did not differ in H_2_O-sprayed and SA-sprayed leaves (Figure 4b); however, at HL, SA-sprayed leaves exhibited increased (7%) efficiency of excitation energy capture by the open PSII rection centers (F*v*′/F*m*′) (Figure 4b). Under MoDS at LL, F*v*′/F*m*′ did not differ in H_2_O-sprayed and SA-sprayed leaves, but at HL, SA-sprayed leaves showed increased (2%) efficiency of excitation energy capture by the open PSII rection centers (F*v*′/F*m*′) (Figure 4b). 

### 2.4. Changes in the Efficiency of the Oxygen Evolving Complex under Normal Growth and Moderate Drought Stress

Under optimal growth conditions, the efficiency of the oxygen evolving complex (OEC, F*v*/F*o*) did not differ in H_2_O-sprayed and SA-sprayed leaves (Figure 5). However, under MoDS, SA-sprayed leaves showed enhanced efficiency (8%) of the OEC (F*v*/F*o*) (Figure 5).

### 2.5. Changes in the Fraction of Closed Photosystem II Reaction Centers, and the Excess Excitation Energy in Photosystem II under Normal Growth and Moderate Drought Stress

The fraction of closed PSII reaction centers (1-*qL*), based on the “lake” model for the photosynthetic unit, under optimal growth conditions (control) did not differ in H_2_O-sprayed and SA-sprayed leaves at LL (Figure 6a); however, at HL, SA-sprayed leaves exhibited a smaller (−9%) fraction of closed PSII reaction centers (1-*qL*) (Figure 6a). Yet, under MoDS, SA-sprayed leaves had a smaller fraction of closed PSII reaction centers (1-*qL*) at both LL (−9%) and HL (−5%) (Figure 6a).

The excess excitation energy (EXC), calculated as (F*v*/F*m* − Φ_PSII_)/F*v*/F*m*, under optimal growth conditions (control), did not differ in H_2_O-sprayed and SA-sprayed leaves at LL (Figure 6b); however, at HL, SA-sprayed leaves exhibited significantly less (−10%) EXC (Figure 6b). Moreover, under MoDS, SA-sprayed leaves presented significantly less EXC at both LL (−5%), and HL (−5%) (Figure 6b).

## 3. Discussion

Climate change’s impacts on agriculture and the increasing world population both threaten global food security [70]. Drought is the main global threat that affects agricultural production [71]. Photosynthesis is the main process in plants that can be intensely disturbed by environmental parameters [72]. Thus, the challenge of improving crop performance by increasing the photosynthetic efficiency of crop plants is a crucial and significant research issue [56,67]. Enhanced photosynthetic efficiency can be accomplished via improved distribution of the absorbed light energy [12]. Absorbed light energy can be used via photochemistry or dissipated via various thermal processes at the light reactions of photosynthesis; these comprise a set of redox reactions which are the basis of energy production in plant cells [23,29,73,74]. When the absorbed light energy exceeds the amount that can be used for photochemistry, increased formation of reactive oxygen species (ROS), such as hydrogen peroxide (H_2_O_2_), superoxide anion radical (O_2_**^•^^−^**), and singlet oxygen (^1^O_2_), occurs [24,31,75,76,77]. Later, ^1^O_2_, is produced from the triplet chlorophyll excited-state (^3^Chl*) which is formed through an intersystem crossing of the singlet excited-state chlorophyl (^1^Chl*) [2,20,31]. Under DS, there is an overexcitation of PSII, because the absorbed light energy exceeds chloroplasts’ capabilities to use it, and the excess photons increase the amount of ^1^Chl* and thus the probability of ^3^Chl* and ^1^O_2_ formation, prompting subsequent photoinhibition [31,38,42,78]. Chlorophyll molecules are the key pigments for capturing light energy and transferring it to the reaction centres and the consequential electron transport in light reactions [20,79,80,81]. 

The decline in chlorophyll content under MoDS in oregano seedlings (Figure 1a) might be attributed to the possible oxidation of chlorophyll molecules [82,83]. However, this reduction in the chlorophyll content under MoDS was partially reversed by the foliar application of SA, which is known to ameliorate oxidative stress and serve as an antioxidant [60,84]. It seems that under MoDS, the application of SA, which acted as an antioxidant, decreased the oxidation of chlorophyll molecules and modulated the chlorophyll content, resulted in improving antenna size. Optimizing antenna size can maximize photosynthetic efficiency [55]. Thus, in SA-sprayed oregano seedlings, the improved antenna size under MoDS growth conditions was followed by an enhancement of PSII photochemistry under both LL and HL. This was evident in the increased Φ*_PSII_* (Figure 2a), the increased q*p* (Figure 3b), the increased ETR (Figure 4a), but also the decreased Φ*_NO_* (Figure 3a) and the decreased EXC (Figure 6b). Using Φ*_NO_*, the probability of ^3^Chl* and ^1^O_2_ formation can be calculated [60,85]. Thus, a decreased Φ*_NO_* reflects the ability of a plant to protect itself against excess light energy that leads to photoinhibition and photodamage [60,86,87,88]. 

The decreased chlorophyll content in oregano leaves under MoDS, compared to no stress, results in the downsizing of their light-harvesting capacity to prevent photo-oxidative stress [53,55,89]. The modulation of antenna size, through foliar application of SA that decreased chlorophyll content (Figure 1a) and enhanced photosynthetic efficiency, was verified under non-stressed conditions and HL. Foliar application of SA, under non-stressed conditions and HL, increased Φ*_PSII_* (Figure 2a), q*p* (Figure 3b), ETR (Figure 4a), and F*v*′/F*m*′ (Figure 4b), and also resulted in less EXC (Figure 6b), a smaller fraction of closed PSII reaction centers (1-*qL*) based on the “lake” model for the photosynthetic unit (Figure 6a), and a significantly lower heat dissipation (Φ*_NPQ_*) (Figure 2b). The significantly lower Φ*_NPQ_*, under non-stressed conditions and HL, after SA application, indicates the photoprotective quality of SA in oregano seedlings against damage by excess illumination [60]. Reducing the size of the light-harvesting antenna has been recognised as an effective approach to mitigate photosynthetic inadequacy related to over-absorption of light energy [90,91].

Limitation of photoprotection under DS subsequently leads to photooxidative damage, indicated by an increase in Φ*_NO_* as well as a decrease in the maximum quantum efficiency of PSII (F*v*/F*m*) [12,39,42,92,93]. Chlorophyll *a* fluorescence analysis revealed a higher value of minimum fluorescence (F*o*) (data not shown), and a significant decrease in F*v*/F*m* (Figure 1b) in both H_2_O-sprayed and SA-sprayed oregano leaves under MoDS. Thus, a higher fraction of absorbed light energy was lost as fluorescence under MoDS compared to optimal growth conditions. Yet, lower F*v*/F*m* values under MoDS (Figure 1b) indicate a higher degree of photoinhibition [94,95]. Nevertheless, SA-sprayed oregano leaves under MoDS had a higher F*v*/F*m* ratio compared to the H_2_O-sprayed leaves (Figure 1b). 

PSII photodamage can appear through photooxidative stress, either at the acceptor side through ^3^Chl*, which by exchanging energy and spinning with O_2_ in the triplet state (molecular oxygen) results in ^1^O_2_ formation, or at the donor side through inactivation of the oxygen-evolving complex (OEC) [60,96,97,98]. Under MoDS, both H_2_O-sprayed and SA-sprayed oregano leaves exhibited a reduced efficiency of the OEC at the donor side (Figure 5). However, SA-sprayed leaves presented enhanced efficiency of the OEC (F*v*/F*o*) compared to those H_2_O-sprayed (Figure 5). The decreased efficiency of the OEC in H_2_O-sprayed oregano seedlings under MoDS (Figure 5) corresponded with a lower F*v*/F*m* ratio (Figure 1b). Drought stress limits the availability of H_2_O for water oxidation, affecting the efficiency of the OEC [99,100]. The higher F*v*/F*m* ratio of SA-sprayed oregano leaves under MoDS compared to that of the H_2_O-sprayed leaves (Figure 1b) was possible due to the amelioration of the oxidative stress, and the decreased quantum yield of non-regulated energy loss (Φ*_NO_*) (Figure 3a), which resulted in decreased ^1^O_2_ formation. Yet, the increased efficiency of the OEC at the donor side (Figure 5), resulted in a significantly lower EXC (Figure 6b), indicating improvements related to PSII efficiency.

The chlorophyll fluorescence parameter 1−q_L_ [101] has been shown to act as a signal to stomatal guard cells [102]. Accordingly, the lower fraction of closed reaction centres, or alternatively, the more oxidized Q*_A_* pool in SA-sprayed leaves under MoDS (Figure 6a), corresponds to a lower stomatal opening, which was accompanied by a lower EXC (Figure 6b), indicating improved PSII efficiency. The fraction of open PSII reaction centers (q_p_) decreases during DS, and this leads to decreases in Φ*_PSII_* and increases in Φ*_NPQ_* [12,92,93,103,104]. However, in SA-sprayed leaves, compared to H_2_O-sprayed, under HL and normal growth conditions or HL and MoDS, the captured light energy was preferentially converted into photochemical energy (Φ*_PSII_*) (Figure 2a), rather than dissipated as heat (Φ*_NPQ_*) (Figure 2b). The enhanced ETR in SA-sprayed leaves compared to H_2_O-sprayed, under MoDS at both LL and HL(Figure 4a) was due to an increased q*p* (Figure 3b) and an increased F*v*′/F*m*′ (Figure 4b). However, SA has been shown to slow down ETR in tobacco [66] but enhance ETR in tomatoes at both LL and HL [60]. In *Hordeum vulgare*, SA triggered a concentration-related decreased efficiency of the OEC, resulting also in a decreased fraction of open PSII centres [105]. It appears that SA’s mode of action depends considerably on several characteristics, such as the plant species, exposure duration, the concentration used, and the environmental conditions [60,61,106,107]. Thus, data on the effects of SA on plant physiological processes under stressed or non-stressed conditions remain debatable [106], but generally it can be recognized that SA has a positive effect on plant responses to many abiotic stresses such as heat, chilling, salinity, drought, and heavy metal toxicity [60,84,108,109,110,111,112,113,114,115,116]. The diverse impact of SA on different plant species may be due to the diversification of the SA signaling and biosynthesis pathways in plants [117].

## 4. Materials and Methods

### 4.1. Plant Material and Growth Conditions

Seedlings of oregano (*Origanum vulgare* L.) were obtained from a plant nursery and transported to a growth chamber with 21 ± 1/19 ± 1 ^o^C day/night temperature, 60 ± 5/70 ± 5% relative humidity day/night, and a 14 h photoperiod, with photosynthetic photon flux density (PPFD) 200 ± 10 μmol photons m^−2^ s^−1^ [60].

### 4.2. Salycilic Acid Treatment

Oregano seedlings under normal growth conditions were sprayed with 1 mM salicylic acid (SA) or double distilled H_2_O, and after 72h the chlorophyll content and PSII function were evaluated [60]. In addition, chlorophyll content and PSII function were evaluated in oregano seedlings that were sprayed with 1 mM SA or double distilled H_2_O, and exposed to moderate drought stress (MoDS). Each plant received 10 mL of 1 mM SA or double distilled H_2_O, applied by a hand sprayer only once during the experiment at 72 h before the measurements. All treatments were performed with four independent biological replicates.

### 4.3. Drought Stress Treatment and Soil Water Status

Moderate drought stress (MoDS) was induced by withholding irrigation of oregano seedlings until a 60% soil volumetric H_2_O content (SWC) was maintained in the control seedlings. SWC was measured with ProCheck device coupled with the soil moisture sensor 5TE (Decagon Devices, Pullman, WA, USA), as described previously [118]. 

### 4.4. Chlorophyll Content

Chlorophyll content was measured photometrically usinga dual wavelength optical absorbance (620 and 920 nm) portable chlorophyll content meter (Model Cl-01, Hansatech Instruments Ltd., Norfolk, UK) [119]. 

### 4.5. Chlorophyll Fluorescence Analysis

Chlorophyll fluorescence analysis of dark-adapted oregano plants was performed as described in detail previously [120], using an Imaging-PAM Fluorometer M-Series MINI-Version (*Heinz Walz GmbH*, Effeltrich, Germany). The minimum (F*o*) and the maximum (F*m*) chlorophyll *a* fluorescence in the dark was measured after 20 min dark adaptation. The maximum chlorophyll *a* fluorescence in the light (F*m*′) was measured after a saturation pulse, while the minimum chlorophyll *a* fluorescence in the light (F*o*′) was computed by Win software (Heinz Walz GmbH, Effeltrich, Germany) as F*o*′ = F*o*/(F*v*/F*m* + F*o*/F*m*′) [121]. Steady-state photosynthesis (F*s*) was measured after 5 min of illumination time with either 205 μmol photons m^−2^ s^−1^, actinic light (AL) low light intensity (LL), which corresponds to the growth light intensity, or with 1000 μmol photons m^−2^ s^−1^, high light intensity (HL). The following chlorophyll fluorescence parameters (Table 1) were estimated by Win software (Heinz Walz GmbH, Effeltrich, Germany).

### 4.6. Statistics

All data were tested for normality with a Shapiro–Wilk test, and for homogeneity of variance with Levene’s test prior to statistical analysis [122]. The populations of variances were not equal, so we performed a Welch’s ANOVA to compare the four treatments, followed by a post hoc analysis with a Games–Howell test [60]. All the analyses were performed in SPSS version 28.0 (IBM, Chicago, IL, United States) for Windows. The data are presented as means ± SD (*n* ≥ 4).

## 5. Conclusions

Salicylic acid application increased the effective quantum yield of PSII photochemistry (Φ*_PSII_*) by enhancing the efficiency of the oxygen evolving complex (OEC) and increasing the fraction of open PSII reaction centers (q*p*), which resulted in an increased electron transport rate (ETR). We can conclude that SA application may reduce the excess excitation energy by reducing ^1^O_2_ formation, and may also enhance the photosynthetic function of oregano seedlings to challenge DS; thus, SA can be regarded as a promising tool for improving the ability of crop plants to face drought episodes in combination with the high light conditions of the Mediterranean area that influence crop production detrimentally. However, since the impact of SA application on different crop plants is diverse, possibly due to the diversification of the SA signaling and biosynthesis pathways in plants, more experiments must be executed in different crop species to establish the large-scale use of SA in agriculture in order to achieve sustainable crop production to confront the challenge of climate change.

## Figures and Tables

**Figure 1 plants-12-00518-f001:**
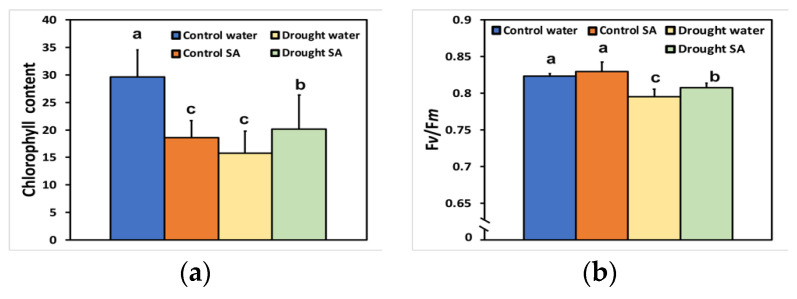
Chlorophyll content (**a**); and maximum efficiency of PSII photochemistry (F*v*/F*m*) (**b**); of oregano seedlings grown under optimal conditions (control) or moderate drought stress (MoDS), and sprayed by 1 mM SA or H_2_O. Error bars represent standard deviations (*n* = 4). Different lowercase letters represent statistical difference (*p* < 0.05).

**Figure 2 plants-12-00518-f002:**
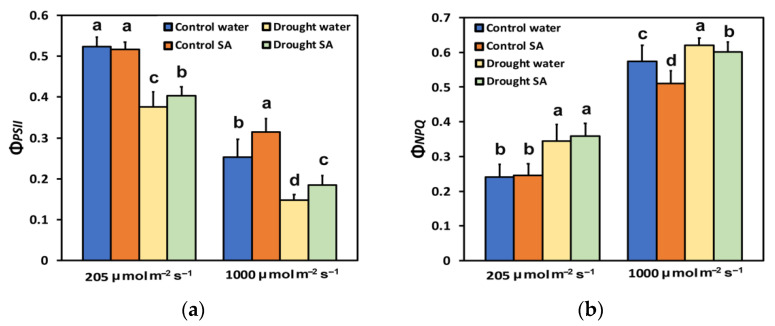
The effective quantum yield of PSII photochemistry (Φ*_PSII_*) (**a**); and the quantum yield of regulated non-photochemical energy loss in PSII (Φ*_NPQ_*) (**b**); of oregano seedlings grown under optimal conditions (control) or moderate drought stress (MoDS), and sprayed by 1 mM SA or H_2_O. Φ*_PSII_* and Φ*_NPQ_* were estimated at 205 (LL) or 1000 (HL) μmol photons m^–2^ s^–1^. Error bars represent standard deviations (*n* = 4). Different lowercase letters, within the same light treatment, represent statistical difference (*p* < 0.05).

**Figure 3 plants-12-00518-f003:**
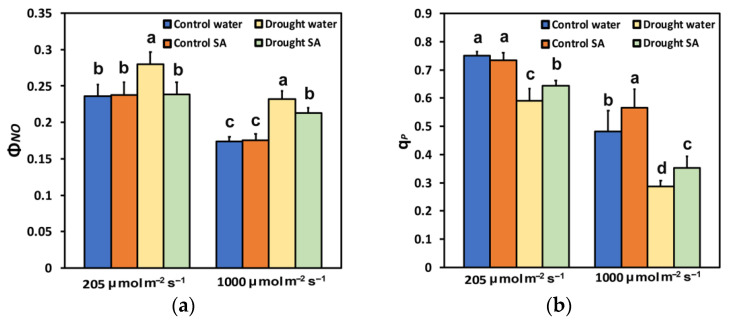
The quantum yield of non-regulated energy loss (Φ*_NO_*) (**a**); and fraction of open PSII reaction centers (q*p*) (**b**); of oregano seedlings grown under optimal conditions (control) or moderate drought stress (MoDS), and sprayed by 1 mM SA or H_2_O. Φ*_NO_* and q*p* were estimated at 205 (LL) or 1000 (HL) μmol photons m^–2^ s^–1^ actinic light (AL) intensity. Error bars represent standard deviations (*n* = 4). Different lowercase letters, within the same light treatment, represent statistical difference (*p* < 0.05).

**Figure 4 plants-12-00518-f004:**
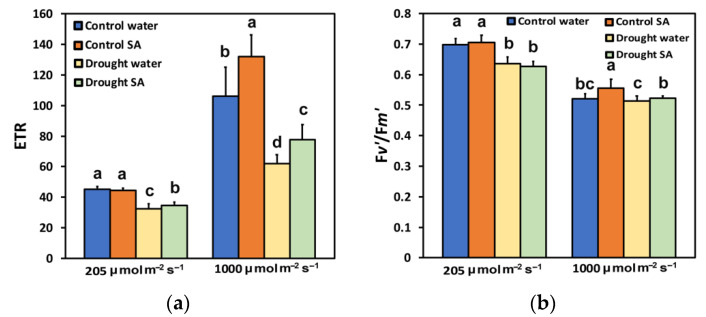
The electron transport rate (ETR) (**a**); and the efficiency of excitation energy capture by the open PSII rection centers (F*v*′/F*m*′) (**b**); of oregano seedlings grown under optimal conditions (control) or moderate drought stress (MoDS), and sprayed by 1 mM SA or H_2_O. ETR and F*v*′/F*m*′ were estimated at 205 (LL) or 1000 (HL) μmol photons m^–2^ s^–1^ actinic light (AL) intensity. Error bars represent standard deviations (*n* = 4). Different lowercase letters, within the same light treatment, represent statistical difference (*p* < 0.05).

**Figure 5 plants-12-00518-f005:**
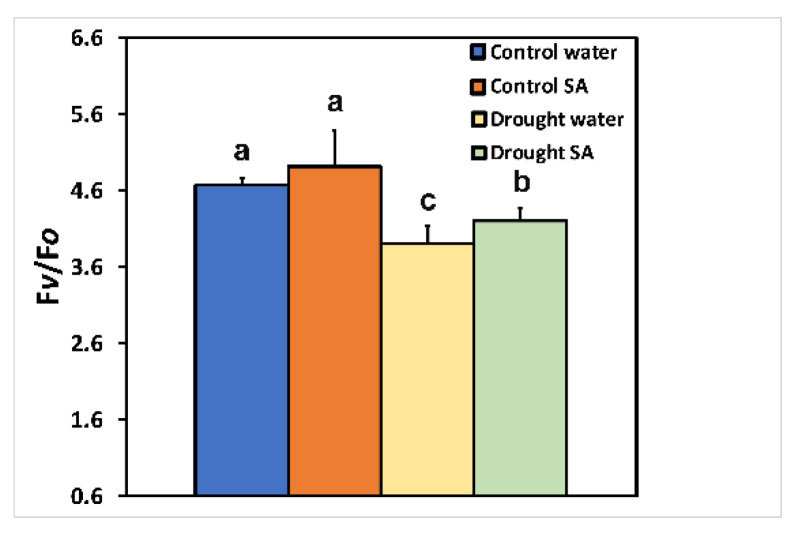
The efficiency of the oxygen evolving complex (OEC, F*v*/F*o*) of oregano seedlings grown under optimal conditions (control) or moderate drought stress (MoDS) and sprayed with 1 mM SA or H_2_O. Error bars represent standard deviations (*n* = 4). Different lowercase letters represent statistical difference (*p* < 0.05).

**Figure 6 plants-12-00518-f006:**
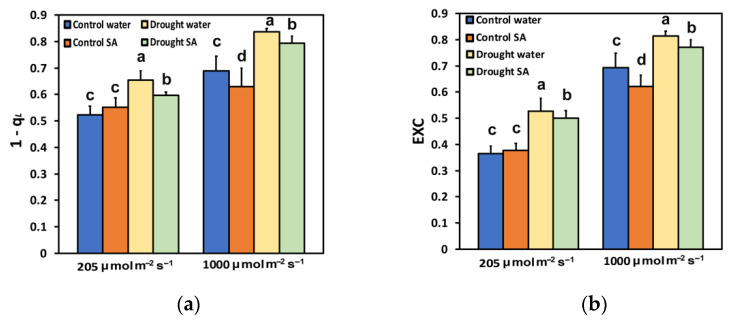
The fraction of closed PSII reaction centers (1-*qL*), based on the “lake” model for the photosynthetic unit (**a**); and the excess excitation energy (EXC) (**b**); of oregano seedlings grown under optimal conditions (control) or moderate drought stress (MoDS) and sprayed with 1 mM SA or H_2_O. 1-*qL* and EXC were estimated at 205 (LL) or 1000 (HL) μmol photons m^–2^ s^–1^ actinic light (AL) intensity. Error bars represent standard deviations (*n* = 4). Different lowercase letters, within the same light treatment, represent statistical difference (*p* < 0.05).

**Table 1 plants-12-00518-t001:** The estimated chlorophyll fluorescence parameters with their definitions and their calculation formulae [adopted from 6].

Parameter	Definition	Calculation	
F*v*/F*m*	Maximum efficiency of PSII photochemistry	(F*m* − F*o*)/F*m*	
Φ*_PSII_*	Effective quantum yield of PSII photochemistry	(F*m*′ − F*s*)/F*m*′	
Φ*_NPQ_*	Quantum yield of regulated non-photochemical energy loss in PSII	F*s*/F*m*′ − F*s*/F*m*	
Φ*_NO_*	Quantum yield of nonregulated energy loss in PSII	F*s*/F*m*	
F*v*′/F*m*′	Efficiency of open PSII centers	(F*m*′ − F*o*′)/F*m*′	
F*v*/F*o*	Efficiency of the oxygen evolving complex (OEC) on the donor side of PSII	(F*m* − F*o*)/F*o*	
ETR	Electron transport rate	Φ_PSII_ × PAR × c × abs, where PAR is the photosynthetically active radiation, c is 0.5, and abs is the total light absorption of the leaf taken as 0.84	
q*p*	Photochemical quenching, representing the fraction of PSII reaction centers in open state (puddle model)	(F*m*′ − F*s*)/(F*m*′ − F*o*′)	
NPQ	Non-photochemical quenching reflecting the dissipation of excitation energy as heat	(F*m* − F*m*′)/F*m*′	
EXC	Excess excitation energy	(F*v*/F*m* − Φ_PSII_)/F*v*/F*m*	
1-q*L*	The fraction of PSII reaction centers in closed state (based on a “lake” model for the photosynthetic unit)	q*p* × F*o*′/F*s*	

## Data Availability

The data presented in this study are available in this article.

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
