# Peer review of "Mechanistic Insights on Salicylic Acid Mediated Enhancement of Photosystem II Function in Oregano Seedlings Subjected to Moderate Drought Stress†"

_plants, 2023, doi:10.3390/plants12030518_

Round 1

Reviewer 1 Report

Review remarks on the manuscript entitled “Mechanistic Insights on Salicylic Acid Mediated Enhancement of Photosystem II Function in Oregano Seedlings Subjected to  Moderate Drought Stress” is submitted by Moustakas et al. to plants journal. The manuscript is well-written and discusses the role of salicylic acid on photosynthetic performance in Oregano seedlings in response to water stress conditions. It’s a great research interest to researchers and academicians.

 The following points must be corrected or described in the details:

·       In the abstract, please mention the % increase data values for more clarity and comparative

·       Please mention the Moderate DS soil moisture water % content level

·       Please improve the language of the Introduction section. Some sentence errors

·       Line 124-125: Why are you using/selecting 1 mM SA level, any specific reason behind this? and why apply only 1 concentration, not more like 2 or more?

·       Line 134: What is the meaning of 72?

·       Line 143-148: Please incorporate % increase or decrease values

·       Discussion section is well organized.

·       Line 348-354: How much water and SA sprayed in a time? and how many times were sprayed during experimentation?

·        Please improve the conclusion section and incorporate significant findings and future recommendations

After incorporation of these changes, the article can be acceptable for publication in the plants journal.

Author Response

·       In the abstract, please mention the % increase data values for more clarity and comparative.

Percentage changes are included in the abstract except from the last general sentence that refers to both low light (LL) and high light (HL) conditions. We did not include there % increases since there were different percentages under LL and HL conditions.

  • Please mention the Moderate DS soil moisture water % content level.

The Moderate DS soil moisture water % is mentioned in Materials and Methods (lines 1106-1107). It was 60 % soil volumetric water content, of control seedlings.

  • Please improve the language of the Introduction section. Some sentence errors.

Sentence errors in Introduction were corrected.

  • Line 124-125: Why are you using/selecting 1 mM SA level, any specific reason behind this? and why apply only 1 concentration, not more like 2 or more?

We used 1 mM SA, because this concentration was found in our previous work (Int. J. Mol. Sci. 2022, 23, 7038) to enhance photosynthesis in tomato plants under normal growth conditions. It was selected after preliminary experiments among three concentrations.

  • Line 134: What is the meaning of 72?

Thank you for pointing this. It is 72 hours (added now, line 187).

  • Line 143-148: Please incorporate % increase or decrease values.

Changes % were incorporated in lines 143-148 (now lines 313-320, the line numbering change is due to an unresolved problem).

  • Discussion section is well organized.

Thank you for your positive comment.

  • Line 348-354: How much water and SA sprayed in a time? and how many times were sprayed during experimentation?

Thank you for your comment. This information was missing. We included in lines 1087-1089 (now) “Each plant received 10 ml of 1 mM SA or double distilled H2O by a hand sprayer only once during the experiment, 72 h before the measurements”.

  • Please improve the conclusion section and incorporate significant findings and future recommendations

Conclusion section was improved.

Reviewer 2 Report

The authors report an interesting study on the protective effects of salicylic acid in oregano crops subjected to water stress. In particular, salicylic acid protects plants from oxidative stress induced by the closure of the stomata generated by drought. Indeed, photosystem II is responsible for the excessive production of ROS and salicylic acid should reduce excessive excitation.

The manuscript is simple, well written, and clear. However there are some critical issues that should be better explained:

1) Lines 101-119, the authors discuss the action of salicylic acid. They mention only a few crops, this part should be enriched with some very recent publications on two other crops.

Particularly:

Yousefvand, P. et al. Salicylic Acid Stimulates Defense Systems in Allium hirtifolium Grown under Water Deficit Stress. Molecules 2022, 27, 3083. doi: 10.3390/molecules27103083;

Biareh, V. et al. Physiological and Qualitative Response of Cucurbita pepo L. to Salicylic Acid under Controlled Water Stress Conditions. Horticulturae 2022, 8, 79. doi: 10.3390/horticulturae8010079

2) The authors should describe the characteristics of the soil used. Indeed, they procure oregano crops in a food market, but do not explain whether the soil is replaced or what its characteristics are.

3) Who carried out the botanical identification of the plants?

4) Are the plants grown in a controlled environment? What time of the year were the measurements taken? Oregano is an aromatic plant that has an interesting volatilome that changes according to months of the year and to abiotic stresses.

5) The figures are too small, they should be redone. The graph legends are illegible.

Author Response

1) Lines 101-119, the authors discuss the action of salicylic acid. They mention only a few crops, this part should be enriched with some very recent publications on two other crops.

Particularly:

Yousefvand, P. et al. Salicylic Acid Stimulates Defense Systems in Allium hirtifolium Grown under Water Deficit Stress. Molecules 2022, 27, 3083. doi: 10.3390/molecules27103083;

Biareh, V. et al. Physiological and Qualitative Response of Cucurbita pepo L. to Salicylic Acid under Controlled Water Stress Conditions. Horticulturae 2022, 8, 79. doi: 10.3390/horticulturae8010079

Thank you for your positive comments. We enriched the manuscript literature by incorporating these two recent publications in the Discussion section (citations 115,116).

2) The authors should describe the characteristics of the soil used. Indeed, they procure oregano crops in a food market, but do not explain whether the soil is replaced or what its characteristics are.

Plants were kept for the period of the experiment in the pots with the soil that were grown in the nursery without any soil replacement.

3) Who carried out the botanical identification of the plants?

The botanical identification of the plants was initially carried out by the plant nursery and verified by the co-author Dr. Ilektra Sperdouli from Institute of Plant Breeding and Genetic Resources, Hellenic Agricultural Organisation–Demeter (ELGO- Demeter).

4) Are the plants grown in a controlled environment? What time of the year were the measurements taken? Oregano is an aromatic plant that has an interesting volatilome that changes according to months of the year and to abiotic stresses.

Plants were kept for the period of the experiment in a growth chamber with 21 ± 1/19 ± 1 oC day/night temperature, 60 ± 5/70 ± 5% relative humidity day/night, and a 14-h photoperiod with photosynthetic photon flux density (PPFD) 200 ± 10 μmol photons m−2 s−1.

5) The figures are too small, they should be redone. The graph legends are illegible.

We enlarged the graph legends in all Figures.

Round 2

Reviewer 2 Report

The authors have improved the manuscript considerably following the suggestions of the Reviewers. In my opinion, the manuscript can now be accepted for publication on PLANTS.